# Significance of Lauren Classification in Patients Undergoing Neoadjuvant/Perioperative Chemotherapy for Locally Advanced Gastric or Gastroesophageal Junction Cancers—Analysis from a Large Single Center Cohort in Germany

**DOI:** 10.3390/cancers13020290

**Published:** 2021-01-14

**Authors:** Rebekka Schirren, Alexander Novotny, Christian Oesterlin, Julia Slotta-Huspenina, Helmut Friess, Daniel Reim

**Affiliations:** 1Department of Surgery, TUM School of Medicine, Technical University Munich, 81675 Munich, Germany; rebekka.schirren@tum.de (R.S.); alexander.novotny@tum.de (A.N.); christian.oesterlin@mri.tum.de (C.O.); helmut.friess@tum.de (H.F.); 2Institute of Pathology, TUM School of Medicine, Technical University Munich, 81675 Munich, Germany; julia.slotta-huspenina@tum.de

**Keywords:** gastric/gastroesophageal cancer, perioperative chemotherapy, Lauren histotype

## Abstract

**Simple Summary:**

Chemotherapy ahead of surgery is standard of care for locally advanced stomach cancer or cancer at the junction between esophagus and stomach in Europe. However, response to chemotherapy may depend on microscopic features of the tumor. Three types were defined before: intestinal, diffuse and mixed types. The authors aimed to investigate if these characteristics influence survival after end of treatment (chemotherapy+surgery) in a large cohort treated in a University hospital. It was found that intestinal type patients demonstrate longer survival after chemotherapy+surgery than those with diffuse types. In the mixed type group no clear conclusion regarding the effect of chemotherapy ahead of surgery may be taken. Conclusively, patients with diffuse type tumors do not benefit from chemotherapy ahead of surgery.

**Abstract:**

Background: the purpose of this analysis was to analyze the outcomes of multimodal treatment that are related to Lauren histotypes in gastro-esophageal cancer (GEC). Methods: patients with GEC between 1986 and 2013 were analyzed. Uni- and multivariate regression analysis were performed to identify predictors for overall survival. Lauren histotype stratified overall survival (OS)-rates were analyzed by the Kaplan–Meier method. Further, propensity score matching (PSM) was performed to balance for confounders. Results: 1290 patients were analyzed. After PSM, the median survival was 32 months for patients undergoing primary surgery (PS) and 43 months for patients undergoing neoadjuvant chemotherapy (nCTx) ahead of surgery. For intestinal types, median survival time was 34 months (PS) vs. 52 months (nCTx+surgery) *p* = 0.07, 36 months (PS) vs. (31) months (nCTx+surgery) in diffuse types (*p* = 0.44) and 31 months (PS) vs. 62 months (nCTx+surgery) for mixed types (*p* = 0.28). Five-/Ten-year survival rates for intestinal, diffuse, and mixed types were 44/29%, 36/17%, and 43/33%, respectively. After PSM, Kaplan–Meier showed a survival benefit for patients undergoing nCTx+surgery in intestinal and mixed types. Conclusion: the Lauren histotype might be predictive for survival outcome in GEC-patients after neoadjuvant/perioperative chemotherapy.

## 1. Introduction

Gastric cancer belongs to the most common malignant diseases worldwide with the highest incidence in Eastern Asia [1]. Despite decreasing incidence in the West, it remains a therapeutic challenge. In the Western hemisphere gastric malignancy is often diagnosed at an advanced stage and, in contrast to Eastern Asia, it is preferably located in the proximal third of the stomach or the gastro-esophageal junction (GEJ) [2]. Hence, multimodal treatment concepts have been introduced after demonstrating outcome benefits in randomized controlled trials [3,4,5]. Nevertheless, not all patients are benefitting from neoadjuvant or perioperative chemotherapy, depending on localization, regimen, and also on the histological subtype. In the past, a signet ring cell, like gastric cancer, was identified to be non-responsive to neoadjuvant chemotherapy [6,7]. However, the data published so far have been difficult to interpret, as there were numerous definitions on the histology of signet ring cell or signet ring cell, like gastric cancer [8]. A pragmatic and feasible sub-classification was only recently published [9]. However, none of the prospective trials investigating the value of neoadjuvant chemotherapy applied this classification system before. Therefore, it is of special interest if already established histopathological classifications may stratify and identify patients to benefit from neoadjuvant/perioperative chemotherapy. This may be accomplished by the widely accepted Lauren classification, because all of the relevant histopathological subtypes (signet ring cell type, poorly-cohesive signet ring cell type, poorly cohesive non-signet-ring cell type, mucinous, papillary, and tubular) are summarized here [10]. Therefore, it was hypothesized that the Lauren histotype dependent histopathologic response may influence survival outcomes after neoadjuvant/perioperative chemotherapy and the aim of this analysis was to evaluate the oncologic outcomes of perioperative/neoadjuvant chemotherapy in a large German single center cohort, depending on the Lauren histotype.

## 2. Results

For this retrospective analysis, the institutional database for gastric cancer patients was screened and identified 2782 patients having been treated by either surgery or chemotherapy followed by surgery. After removing all cases not fulfilling the defined inclusion criteria (n = 1573), 1209 patients were included in this analysis. 730 patients underwent primary surgery and 479 underwent neoadjuvant/perioperative chemotherapy ahead of surgery. Overall, 663 were diagnosed with Lauren intestinal (398 surgery, 265 nCTx), 359 Lauren diffuse (216 surgery, 143 nCTx), and 187 Lauren mixed type (116 surgery, 71 nCTx). In the entire patient cohort, 247 patients received PLF (20.4%), 41 patients PLF+Taxol (3.4%), 53 (4.4%) OLF, 47 (3.9%) MAGIC, 17 (1.4%) FLOT, and 63 patients received modified regimens (5.2%). The analysis of the baseline characteristics showed significant differences between the primary surgery and neoadjuvant/perioperative chemotherapy group regarding gender distribution (more female patients for intestinal and diffuse. but not mixed Laurentype), older age for patients undergoing primary surgery (all Lauren subtypes), higher proportion of distal cancer locations in primary surgery patients (all groups, especially intestinal type), less advanced cT-stages in the surgery only group (cT2 vs. cT3/4 over all Lauren subtypes), earlier clinical stages, higher proportion of patients requiring extension to the distal esophagus in the chemotherapy group (all Lauren subtypes), higher D2 rates and higher median number of dissected lymph-nodes (LN) in patients undergoing direct surgery (especially in intestinal and mixed Lauren histotype, not so in diffuse type), more pT4a cancers in the primary resection group for all of the Lauren subtypes, earlier UICC stages in those patients undergoing neoadjuvant/perioperative chemotherapy. The proportion of Lauren subtypes, histiopathologic grading, R0-status, and complication rates were balanced between the groups. The histopathologic response rates (Becker Ia/Ib) were 22% in Lauren intestinal type, 9% in Lauren diffuse type, and 21% in Lauren mixed type tumors. Table 1, Table 2 and Table 3 depict the complete baseline characteristics.

The median follow-up was 30 months (range 1–242 months), comprising of 61 months (range 1–242 months) for survivors and 19 months (range 1–183) months for deceased patients. During the follow-up period 658 patients (54.4%) died, the five-year survival rate was 42%, the ten-year survival rate was 32%. Median survival was 38 months for patients undergoing primary surgery and 46 months for patients undergoing chemotherapy ahead of surgery (*p* = 0.06). Five-year survival rates (FYSR)/ten-year survival rates (TYSR) after primary surgery and after neoadjuvant/perioperative chemotherapy followed by surgery were identical (44/33%). The UICC stage dependent analysis revealed that this effect was only reproducible in UICC stage III (19 vs. 24 months median survival, *p* = 0.03), but not in the other UICC stages (UICC I: median survival not met, *p* = 0.58; UICC II: median survival 72 (surgery) vs. 57 (nCTx+surgery) months, *p* = 0.7). In patients with Lauren intestinal subtype, the median survival time was 51 months (45 months for primary surgery vs. 57 months for neoadjuvant/perioperative chemotherapy + surgery, *p* = 0.025), in the diffuse type group 33 months (35 months for primary surgery vs. 28 months for neoadjuvant/perioperative chemotherapy + surgery, *p* = 0.16) and 40 months (26 months for primary surgery vs. 62 months for neoadjuvant/perioperative chemotherapy + surgery, *p* = 0.05) in the Lauren mixed type group. FYSR and TYSR for patients with Lauren intestinal, diffuse, and mixed subtype were 48/35%, 39/31%, and 42/32%, respectively.

The following variables were included in the cox regression analysis: age, gender, localization, neoadjuvant/perioperative chemotherapy, UICC-stage, Lauren subtype, number of dissected LN, R-stage, grading, and postoperative complications, because these are the most relevant factors in predicting survival. The pT- and pN-stages were not included, as these factors are summarized in the UICC-stage. All of the factors were entered in the multivariate model without selection. Univariate regression analysis revealed age, tumor location (all locations), all UICC-stages, Lauren intestinal and diffuse subtypes, R-status, grading, and the occurrence of postoperative complications to be significantly associated with post-therapeutic survival (Table 4). The multivariate analysis demonstrated that age, localization (proximal, middle, distal), application of neoadjuvant/chemotherapy, all UICC-stages, all Lauren subtypes, R-stage, and occurrence of postoperative complications were significantly and independently related to postoperative survival. Because of the imbalanced baseline characteristics, propensity score matching (PSM) was performed, and further analysis was performed on the PS-matched cohorts.

### Results after PSM

Those variables demonstrating clinically meaningful baseline differences within the respective Lauren subgroups were matched through PSM (age, gender, location, clinical stage, UICC stage) in order to balance possible confounders (Appendix A). The matching algorithm matched 170 patients each (surgery/nCTx+surgery) in the Lauren intestinal, 105 patients each in the Lauren diffuse, and 56 patients each in the Lauren mixed subtype groups. Analysis of the baseline characteristics demonstrated that the following variables were then well balanced in all of the groups: Gender, age distribution, tumor localization, clinical stages, D2 dissection rate, number of dissected LN, postop complications, pT-stages (in the Lauren intestinal and mixed, not in the diffuse subtype), UICC (intestinal and mixed subtypes, not diffuse), and grading and R0 status. Table 1, Table 2 and Table 3 show the results.

Median follow-up was 26 months (range 1–242 months), comprising of 55 months (range 1–242 months for survivors and 16 months (range 1–144) months for deceased patients. During the follow-up period, 367 patients (55.4%) died, the five-year survival rate was 41%, and the ten-year survival rate was 25%. The median survival was 32 months for patients undergoing primary surgery and 43 months for patients undergoing chemotherapy ahead of surgery (*p* = 0.16). FYSR/TYSR after primary surgery and after neoadjuvant/perioperative chemotherapy, followed by surgery were 42/31% and 44/32%. The UICC stage dependent analysis revealed no significant survival differences for UICC stages I and II (UICC I: median survival not met, *p* = 0.33; UICC II: median survival 91 (surgery) vs. 80 (nCTx+surgery) months, *p* = 0.72). In UICC III, there was a statistically significant survival difference in favor of those patients undergoing neoadjuvant/perioperative chemotherapy (median survival 18 (surgery) vs. 26 (nCTx+surgery) months (*p* = 0.02), Appendix A. In patients with Lauren intestinal subtype, the median survival time was 46 months (34 months for primary surgery vs. 52 months for neoadjuvant/perioperative chemotherapy + surgery, *p* = 0.07), in patients with diffuse subtype group 35 months (36 months for primary surgery vs. 31 months for neoadjuvant/perioperative chemotherapy + surgery, *p* = 0.44) and 57 months (31 months for primary surgery vs. 62 months for neoadjuvant/perioperative chemotherapy + surgery, *p* = 0.28) in patients with the Lauren mixed subtype. FYSR/TYSR for Lauren intestinal, diffuse, and mixed subtypes were 44/29%, 36/17%, and 43/33%, respectively. Kaplan–Meier analysis revealed that survival benefit for those patients undergoing neoadjuvant/perioperative chemotherapy was detectable for Lauren intestinal (*p* = 0.07) and mixed types (0.28) without statistical significance (Figure 1 and Figure 2). The overall survival was (statistically not significantly) worse for Lauren diffuse type gastric cancer patients when undergoing neoadjuvant/perioperative chemotherapy (*p* = 0.44), (Figure 3). A survival benefit was detectable for Lauren intestinal type patients revealing histopathologic response (HPR) (median survival unmet vs. 43 months in non-responders and 34 months in patients undergoing primary surgery, *p* = 0.01) (Figure 4). There was no significant survival difference between patients undergoing primary surgery and non-responders to nCTx (*p* = 0.65) (Figure 4). This was not reproducible in Lauren diffuse type patients: The median survival was 21 months in responders vs. 33 months in non-responders (*p* = 0.52) and 36 months in patients undergoing primary surgery (*p* = 0.49). There was no survival difference in patients undergoing primary surgery and non-responders to nCTx (*p* = 0.5) (Figure 5). In the Lauren mixed type patients, there was a trend towards improved survival for responders without statistical significance: the median survival was 103 months in responders vs. 57 months in non-responders (*p* = 0.12) and 31 months in patients undergoing primary surgery (*p* = 0.13). There was no survival difference in patients undergoing primary surgery and non-responders to nCTx (*p* = 0.5) (Figure 6).

Recurrence rates and disease free survival were analyzed for the PS-matched groups. In the intestinal type subgroup, the recurrence rates were 79/170 (46.5%) in the surgery only group as compared to 89/170 (52.4%) in the chemotherapy + surgery group (*p* = 0.33). Disease free median survival was 30 months (1–176) in the primary surgery group and 29.5 months (1–242) in the chemotherapy + surgery group (HR 1.12; CI95% 0.83–1.12; *p* = 0.45). In the diffuse type subgroup, the recurrence rates were 62/105 (59%) in the surgery only group compared to 67/105 (63.8%) in the chemotherapy + surgery group (*p* = 0.57). Disease free median survival was 24 months (1–176) in the primary surgery group and 17 months (1–204) in the chemotherapy+surgery group (HR 1.28; CI95% 0.91–1.81; *p* = 0.16). In the mixed type subgroup, the recurrence rates were 35/56 (62.5%) in the surgery only group when compared to 25/56 (44.6%) in the chemotherapy + surgery group (*p* = 0.09). The disease free median survival was 15.5 months (1–171) in the primary surgery group and 38 months (3–202) in the chemotherapy+surgery group (HR 0.52; CI95% 0.31–0.87; *p* = 0.01).

## 3. Discussion

This analysis of a large single center cohort, including 1209 patients, demonstrates an association between the benefit of neoadjuvant chemotherapy and the Lauren subtype. Based on the presented data, only those patients demonstrating the intestinal subtype benefit from the application of neoadjuvant chemotherapy for locally advanced gastric cancer. However, this only holds true for those patients developing histopathologic regression. In contrast, there was no benefit for those patients with diffuse subtype. In the diffuse type group, those patients undergoing neoadjuvant chemotherapy even demonstrated a deterioration of survival when compared to patients who had primary surgery. Patients with Lauren mixed type features revealed a potential benefit, especially those responding to chemotherapy; however, this was not statistically significant. Neoadjuvant/perioperative chemotherapy has become standard of care in Europe and it has become manifest in most of the guidelines for locally advanced gastric and gastroesophageal junction cancers [3,11,12,13]. However, in recent years, it has become increasingly clear that chemotherapy may not be effective for all patients in the same manner. The overall survival rates still range between 20–40% after five years [7,11,14,15]. This analysis surprisingly demonstrates that patients having undergone surgery only revealed survival rates around 40%, and this was improved to over 70% when intestinal subtype tumors demonstrated good histopathologic response. In many studies, the histological subtype was described as an independent factor of survival [15,16,17,18], but it also determines the effectiveness of the chemotherapy administered. However, the histological subtypes are so far not sufficiently respected in regard of therapeutic decisions, for which the clinical tumor stage is still the only tool to be applied when multimodal therapies are recommended. This is underlined by the present analysis, in which patients were subjected to neoadjuvant/multimodal chemotherapy without respect to the Lauren subtype. Taking into account that only 49 of 331 patients (15%) demonstrated real benefit from preoperative chemotherapy, it has to be stated that 85% of the patients were treated ineffectively and may even have been harmed by (ineffective) cytotoxic drugs (Appendix A). The Spanish AGAMENON research group already published a correlation between the response to chemotherapy and the Lauren subtype in 2017. They also pointed out that there were no subgroup analyses in the large therapy trials for locally advanced stomach cancer, although there were indications of a link [7]. Further, the AGAMENON study incorporated almost only patients undergoing treatment for metastatic disease, which is not comparable to the present analysis. Another important analysis was the multicentric retrospective FREGAT study, which analyzed a similar cohort [19]. However, the French analysis was related to the impact of signet ring cell differentiation on oncologic outcomes and not exactly to Lauren diffuse types. In the present analysis it was found that there was not a single Lauren diffuse type cancer without signet ring cells. None the less, Lauren diffuse types should not be equalized to signet ring cell differentiation. An international European group proposed a new definition for signet ring cell containing gastric cancer, as this is still a matter of debate between surgeons, oncologists, and pathologists [9]. However, this new consensus is neither ratified nor prospectively analyzed nor evaluated in patients undergoing neoadjuvant/multimodal chemotherapy. Biological and prognostic differences for gastric cancer are difficult to evaluate in studies due to the fact that there are different histological classifications for gastric carcinoma histological phenotypes and there is no uniform classification. This is considerably relevant when new chemotherapeutic regimens (FLOT) are propagated as effective in signet ring cell gastric cancer. The prospective FREGAT study (PRODIGE-19-FFCD1103-ADCI002) is currently the only trial that is going to elucidate whether direct surgery is a potential option for signet ring cell type gastric cancer [20]. Another factor that does not allow for direct comparisons is the issue that there is no broad consensus on which tumor regression classification to use (Becker, Mandard, Cologne, etc.). Certainly, this analysis should be replicated by different centers, applying different tumor regression systems in order to determine whether the Lauren diffuse subtype is a non-responding entity. Therefore, the aim of future studies should be to unify the different histological classifications for gastric cancer in order to further investigate the influence of histology on survival and prognosis.

The limitations of this analysis are certainly the monocentric character of the study, the long observation period during which both surgical and perioperative regimens have changed, different chemotherapy regimens having been used, and the fact that FLOT, as the current standard of care, is still underrepresented in this analysis. Although potential biases inherent to the Lauren type subgroups were possibly corrected by PSM, this method cannot compensate for unconscious and biological biases or those factors not having been determined. More than that, it is critical that the PSM resulted in a relatively small number of patients per group. Therefore, no exact statements can be made about unmatched patients. Another limitation is that the PS-matching did not balance adequately for the UICC stages in the Lauren diffuse type subgroup, so the balance is skewed towards more advanced cases in the nCTx+surgery group, which might limit further conclusions regarding survival prognosis. Further, the generalizability of the present results is certainly restricted, as the practice of neoadjuvant/multimodal chemotherapy is only evident in the Western (mostly European) world and the findings are not transferable to Asian patients, due to ethnicity and, more importantly, due to the fact that preoperative chemotherapy is not a standard of care in countries, such as Korea, Japan, and China.

Molecular markers, including microsatellite instability (MSI-H) and the Cancer Genome Atlas (TCGA) molecular subtypes [21], which could have influenced the results of our study were not assessed and is a limitation of our study. A recently published meta-analysis demonstrated that MSI-H could predict outcomes to neoadjuvant chemotherapy [22]. However, this same meta-analysis revealed that MSI-H comprised a very small proportion (2.4%) of non-intestinal type gastric cancers. In the present analysis the number of patients not responding to chemotherapy in the diffuse type group was markedly higher (>80% for non-intestinal type cancer), which does not explain the influence of MSI-status only. Beyond molecular factors, the amount of chemotherapy administered may have also been a confounding factor. Although most of the patients received neoadjuvant chemotherapy only (94.6%), relatively few patients (5.4%) received the perioperative FLOT/MAGIC regimens (i.e., pre-operative + post-operative). We are unable to determine the influence of post-operative chemotherapy on the outcomes in our study due to incomplete data available about the administration of post-operative chemotherapy. Nevertheless, despite these limitations, the results of our study raise questions regarding the benefit of neoadjuvant/perioperative chemotherapy in diffuse-type gastric cancer. Complete surgical resection remains the only curative option for gastric cancer patients, even if overall survival is markedly shorter for patients with diffuse-type histology as compared to intestinal or mixed type histologies. To the authors’ knowledge, except for possibly MSI status, which represents a small proportion of patients, there is no existing biomarker in clinical practice that can adequately predict clinical and histopathologic response to neoadjuvant chemotherapy, and future research on identifying other molecular markers are needed.

The clinical implications of this analysis would be to carefully evaluate the application of neoadjuvant/perioperative chemotherapy in diffuse type patients. It remains speculative as to whether a multimodal treatment concept would be harmful for those patients, but it was demonstrated here that it is not beneficial either. Certainly, R0-surgery remains the only option of curation in these patients, even if the overall survival is markedly shorter than in intestinal or mixed type patients.

## 4. Materials and Methods

### 4.1. Patients

Data from patients who underwent curative surgery for gastroesophageal cancer at the Surgical Department of TUM School of Medicine from 1987 to 2017 were extracted from a prospectively documented database. The data were obtained from the medical records and then transferred to the institutional databases as soon as the patients were discharged from inpatient hospital care. The inclusion criteria for this analysis were: histologically proven gastroesophageal cancer (Siewert type II/III, all non-cardia cancers) staged cT2-cT4cN_any_ undergoing neoadjuvant/perioperative chemotherapy after multidisciplinary team review, the Lauren histotype confirmed by expert pathologist (intestinal, diffuse or mixed type). The exclusion criteria were: Siewert type I, metastatic disease, hospital mortality within 30 days, loss of follow-up within a 60 months period, macroscopic residual cancer after surgery (R2), and indeterminate Lauren histotypes. Neoadjuvant/perioperative treatment consisted of either preoperative two cycles cisplatin or oxaliplatin/leucovorin/5-FU (PLF/OLF) only or perioperative three cycles of ECX/ECF (MAGIC) or perioperative four cycles FLOT [12,23]. All of the surgical procedures were performed according to the Japanese guidelines for GC treatment, including standardized D2-lymphnode dissection [24]. In the case of GE junction cancer (Siewert type II and III), the surgical procedure was extended to the distal esophagus. All of the patients received intraoperative frozen sections for the oral resection margin in order to confirm R0 resection. If the resection margin was positive, the surgical procedure was extended to the distal esophagus and esophagectomy was carried out whenever necessary. Circumferential and aboral resection margins were not determined intraoperatively on a routine basis. All of the resected specimens were examined by one or two specialized pathologists, being classified according to the TNM-classification and staged according to UICC-recommendations (8th edition) [25]. The histopathologic response was graded according to the Becker classification. Patients with 0–10% remnant viable tumor cells within the tumor area were graded as histopathologic responders (Becker Ia/Ib), whereas all other patients (Becker II (10–50% remnant viable tumor cells) and Becker III (>50% remnant viable tumor cells)) were graded as histopathologic non-responders [26]. Except for patients receiving FLOT or MAGIC regimens (n = 64, 5.4%, adjuvant chemotherapy was not considered on a routine basis. Following oncologic surgery, all of the patients were followed up every six to twelve months in an outpatient department (Roman Herzog Comprehensive Cancer Center) over the next five years by EGD and CT scans according to the institutional protocol.

Only deceased or surviving patients with a complete follow-up of at least 60 months were included in this analysis. Survival was computed from the day of surgery. The dataset consisted of patients’ gender, age, location (upper, middle, lower third), clinical stages (cT2N0, cT1/cT2cN+, cT3/cT4cN0, cT3/cT4N+), number of dissected lymph nodes, postoperative complications (none, Clavien–Dindo Grade I/II and III/IV), pT- (pT1/pT2/pT3/pT4), pN-(pN0/pN1/pN2/pN3), and UICC-stages (UICC-I/-II/-III), grading (G1/2, G3/4), R-status (R0/R1), Lauren subtype (intestinal, diffuse, mixed), and follow-up period with survival status. Institutional Review Board (IRB)-approval for this study was obtained according to local guidelines (IRB Registration: 364/20 S).

### 4.2. Statistical Analysis

Descriptive statistics on demographic and clinical tumor characteristics were calculated as the mean ± standard deviation (continuous variables) and frequencies (categorical variables). The survival time was calculated from the day of surgery to death or last follow up date (at least 60 months after surgery for survivors). The Kaplan–Meier method was used in order to estimate the survival probabilities stratified by the application of neoadjuvant/perioperative chemotherapy. The log-rank test was used to compare the estimated survival. Survival prognosticators were analyzed by uni- and multivariate cox regression analyses. The variables that entered into the model were age, tumor location (all locations), all UICC-stages, Lauren intestinal and diffuse subtypes, R-status, grading, and occurrence of postoperative complications. After univariate analysis, all of the variables were entered in the multivariate model. Statistical analyses were performed while using SPSS version 25 (IBM Inc., Ehningen, Germany). PSM was performed with R and the MatchIt Plugin (Version 3.01, Vienna, Austria, URL http://www.R-project.org/). *p*-values of less than 0.05 were considered to be statistically significant. This retrospective analysis was approved by the local IRB (No.364/20s; Ethikkommission der Fakultät für Medizin, TUM School of Medicine).

## 5. Conclusions

In conclusion, the present findings demonstrate that the Lauren subtype might be a relevant prognostic factor in relation to overall survival after neoadjuvant/perioperative chemotherapy for locally advanced gastric or gastroesophageal cancer. Data from this analysis suggest that patients with a diffuse subtype may not benefit from neoadjuvant chemotherapy, but further exploration of other factors (e.g., molecular markers, MSI status, EBV-status, etc.), validation in prospective studies, and evaluation of other novel treatments (e.g., immune checkpoint inhibitors) are urgently required.

## Figures and Tables

**Figure 1 cancers-13-00290-f001:**
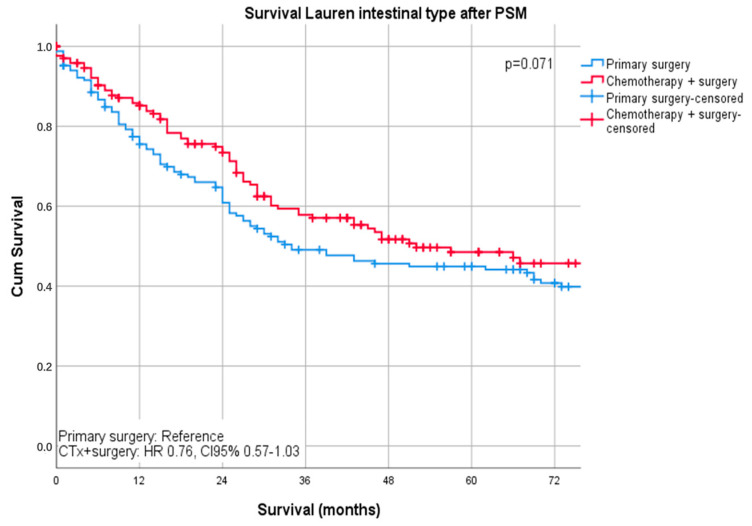
Survival curves for Lauren intestinal subtype after PSM stratified by surgery only vs. chemotherapy plus surgery.

**Figure 2 cancers-13-00290-f002:**
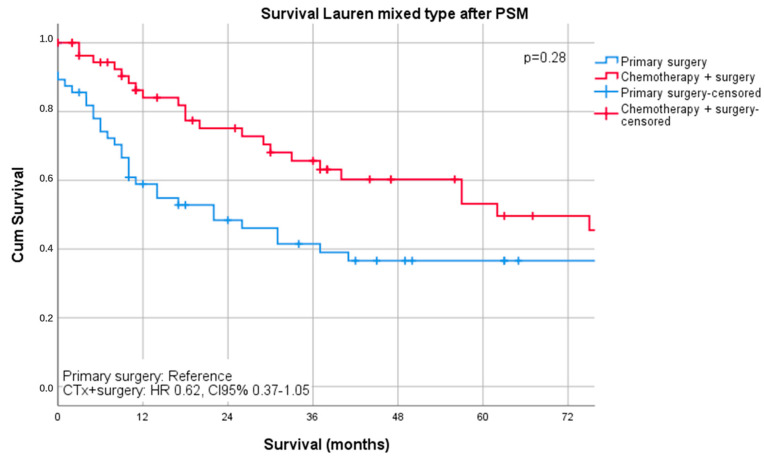
Survival curves for Lauren mixed subtype after PSM stratified by surgery only vs. chemotherapy plus surgery.

**Figure 3 cancers-13-00290-f003:**
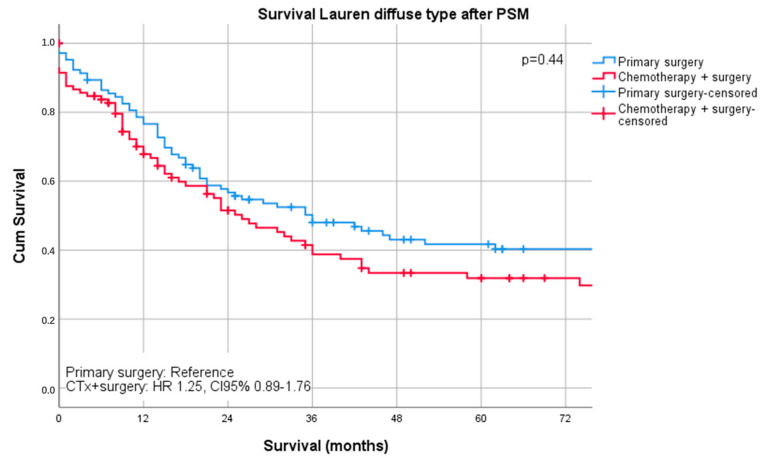
Survival curves for Lauren diffuse subtype after PSM stratified by surgery only vs. chemotherapy plus surgery.

**Figure 4 cancers-13-00290-f004:**
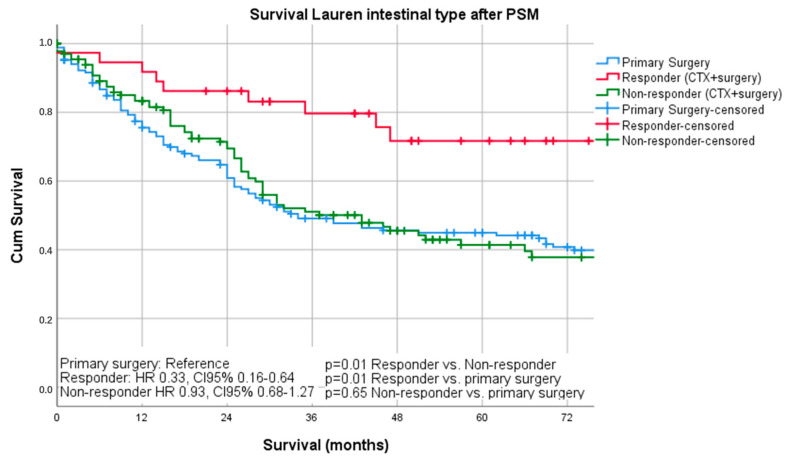
Survival curves for intestinal subtype after PSM differentiated by responders, non-responders, and surgery only.

**Figure 5 cancers-13-00290-f005:**
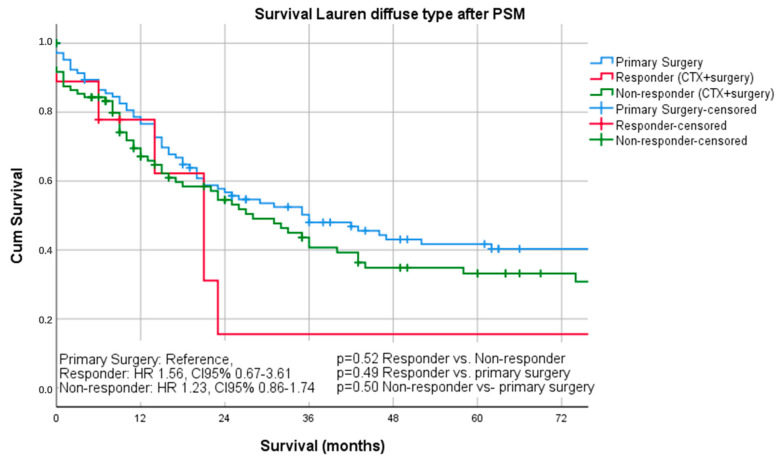
Survival curves for diffuse subtype after PSM differentiated by responders, non- responders and surgery only.

**Figure 6 cancers-13-00290-f006:**
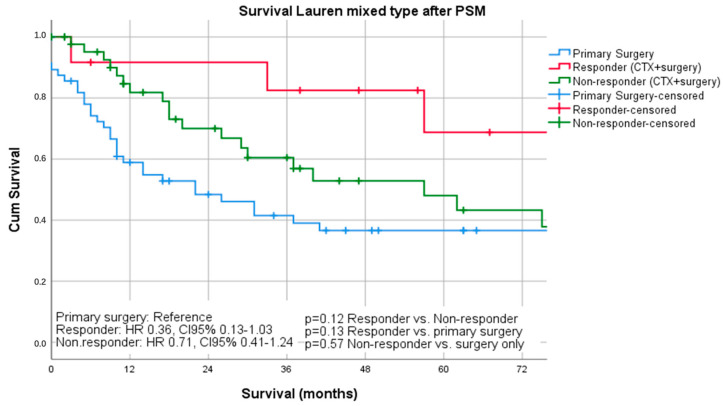
Survival curves for mixed subtype after PSM differentiated by responders, non-responders and surgery only.

**Table 1 cancers-13-00290-t001:** Baseline characteristics for patients with intestinal Lauren subtype before and after propensity score matching (PSM).

	Intestinal Subtype (n = 663), Unmatched					Intestinal Subtype (n = 340) PS-Matched				
	Surgery Only (n = 398)		CTX + Surgery (n = 265)		*p*-Value	Surgery Only (n = 170)		CTX + Surgery (n = 170)		*p*-Value
	n	%	n	%		n	%	n	%	
Gender					**<0.001**					0.14
Female	132	33.17	42	15.85		30	17.65	42	24.71	
Male	266	66.83	223	84.15		140	82.35	128	75.29	
Age	68.7 ± 10.8		60.1 ± 10.5		**<0.001**	65.8 ±10.3		61.6 ±11.1		**<0.001**
<70	188	47.24	206	77.74	**<0.001**	112	65.88	114	67.06	0.91
>70	210	52.76	59	22.26		58	34.12	56	32.94	
Localization					**<0.001**					0.24
Proximal	238	59.80	219	82.64		129	75.88	134	78.82	
Middle	63	15.83	21	7.92		15	8.82	16	9.41	
Distal	90	22.61	25	9.43		22	12.94	20	11.76	
Total	7	1.76	0	0.00		4	2.35	0	0.00	
Clinical Staging					**<0.001**					0.21
cT2 cN+/cNx	161	40.45	31	11.70	**<0.001**	30	17.65	31	18.24	0.99
cT3/cT4 cN0	23	5.78	26	9.81		15	8.82	15	8.82	
cT3/cT4 cN+/cNx	213	53.52	208	78.49		125	73.53	124	72.94	
Dissected LN (Median)	33 (1–105)		29 (5–71)		**0.002**	33 (7–105)		30 (12–71)		0.10
≤25	101	25.38	97	36.60		44	25.88	51	30.00	0.47
>25	297	74.62	168	63.40		126	74.12	119	70.00	
Complications					0.25					0.58
None	298	74.87	187	70.57		123	72.35	125	73.53	
CD I/II	67	16.83	46	17.36		30	17.65	24	14.12	
CD III-V	33	8.29	32	12.08		17	10.00	21	12.35	
pT					**<0.001**					0.11
pT0/is	0	0.00	14	5.28		0	0.00	6	3.53	
pT1a	10	2.51	5	1.89		1	0.59	2	1.18	
pT1b	35	8.79	21	7.92		5	2.94	7	4.12	
pT2	66	16.58	42	15.85		24	14.12	24	14.12	
pT3	183	45.98	139	52.5		92	54.12	99	58.24	
pT4a	84.00	21.11	38	14.3		40.00	23.53	28	16.47	
pT4b	14	3.52	6	2.26		8	4.71	4	2.35	
pN					0.65					0.27
pN0	141	35.43	104	39.25		55	32.35	53	31.18	
pN1	85	21.36	54	20.4		34	20.00	36	21.18	
pN2	74	18.59	51	19.2		33	19.41	38	22.35	
pN3a	70	17.59	44	16.6		28	16.47	34	20.00	
pN3b	28	7.04	12	4.53		20	11.76	9	5.29	
UICC					**0.001**					0.19
UICC 0	0	0.00	14	5.28		0	0.00	6	3.53	
UICC IA	34	8.54	18	6.79		5	2.94	4	2.35	
UICC IB	37	9.30	32	12.1		15	8.82	13	7.65	
UICC IIA	77	19.35	44	16.6		33	19.41	33	19.41	
UICC IIB	71	17.84	49	18.5		28	16.47	32	18.82	
UICC IIIA	83	20.85	50	18.9		38	22.35	38	22.35	
UICC IIIB	68	17.09	45	17		31	18.24	34	20.00	
UICC IIIC	28	7.04	13	4.91		20	11.76	10	5.88	
Grading					0.17					0.66
G1/G2	164	41.21	124	46.79		72	42.35	77	45.29	
G3/G4	234	58.79	141	53.21		98	57.65	93	54.71	
R					0.08					0.86
R0	373	93.72	238	89.81		152	89.41	154	90.59	
R1	25	6.28	27	10.19		18	10.59	16	9.41	
Tumor regression grade										
Becker Ia/Ib			70	26.42				37	21.76	
Becker II			66	24.91				43	25.29	
Becker III			129	48.68				90	52.94	

Legend: cT1 = Mucosa/Submucosa; cT2 = Muscularis propria; cT3 = Serosa; cT4 = Adjacent organs; cN0 = no lymph nodemetastasis detected during staging, cN+ = locoregional lymph node metastasis evident during staging; CD = Clavien Dindo Classification; Staging according to UICC 8th edition; *p*-values printed in bold are considered statistically significant.

**Table 2 cancers-13-00290-t002:** Baseline characteristics for patients with diffuse Lauren subtype before and after PSM.

	Diffuse Subtype (n = 359), Unmatched					Diffuse Subtype (n = 210) PS-Matched				
	Surgery Only (n = 216)		CTX + Surgery (n = 143)		*p*-Value	Surgery Only (n = 105)		CTX + Surgery (n = 105)		*p*-Value
	n	%	n	%		n	%	n	%	
Gender					**0.004**					1.00
Female	114	52.78	53	37.06		41	39.05	42	40.00	
Male	102	47.22	90	62.94		64	60.95	63	60.00	
Age	63.9 ±12.3		56.2 ±11.9		**<0.001**	60.8 ±12.1		57.1 ±12.4		**0.03**
<70	138	63.89	123	86.01	**<0.001**	84	80.00	85	80.95	1.00
>70	78	36.11	20	13.99		21	20.00	20	19.05	
Localization					**0.03**					0.14
Proximal	74	34.26	63	44.06		40	38.10	49	46.67	
Middle	63	29.17	35	24.48		30	28.57	24	22.86	
Distal	63	29.17	27	18.88		30	28.57	21	20.00	
Total	16	7.41	18	12.59		5	4.76	11	10.48	
Clinical Staging					**<0.001**					0.40
cT2 cN+/cNx	75	34.72	14	9.79	**<0.001**	15	14.29	14	13.33	
cT3/cT4 cN0	19	8.80	21	14.69		12	11.43	12	11.43	
cT3/cT4 cN+/cNx	122	56.48	108	75.52		78	74.29	79	75.24	
Dissected LN (Median)	34 (1–104)		30 (9–89)		**0.01**	35 (1–102)		31 (9–70)		0.13
≤25	55	25.46	36	25.17	1.00	26	24.76	23	21.90	0.74
>25	161	74.54	107	74.83		79	75.24	82	78.10	
Complications					0.77					0.4
None	160	74.07	107	74.83		81	77.14	76	72.38	
CD I/II	30	13.89	22	15.38		13	12.38	20	19.05	
CD III-V	26	12.04	14	9.79		11	10.48	9	8.57	
pT					**<0.001**					**0.002**
pT0/is	0	0.00	3	2.10		0	0.00	1	0.95	
pT1a	18	8.33	1	0.70		6	5.71	0	0.00	
pT1b	18	8.33	6	4.20		8	7.62	2	1.90	
pT2	16	7.41	8	5.59		4	3.81	4	3.81	
pT3	48	22.22	65	45.5		22	20.95	46	43.81	
pT4a	104.00	48.15	54	37.8		57	54.29	46	43.81	
pT4b	12	5.56	6	4.20		8	7.62	6	5.71	
pN					**0.05**	35	33.33	29	27.62	**0.03**
pN0	76	35.19	61	42.66		8	7.62	17	16.19	
pN1	23	10.65	21	14.7		23	21.90	20	19.05	
pN2	42	19.44	22	15.4		18	17.14	29	27.62	
pN3a	39	18.06	29	20.3		21	20.00	10	9.52	
pN3b	36	16.67	10	6.99						
UICC					**<0.001**					**0.006**
UICC 0	0	0.00	3	2.1		0	0.00	1	0.95	
UICC IA	29	13.43	6	4.2		12	11.43	1	0.95	
UICC IB	11	5.09	4	2.8		3	2.86	1	0.95	
UICC IIA	22	10.19	33	23.1		9	8.57	17	16.19	
UICC IIB	30	13.89	26	18.2		14	13.33	15	14.29	
UICC IIIA	48	22.22	33	23.1		28	26.67	32	30.48	
UICC IIIB	39	18.06	26	18.2		16	15.24	26	24.76	
UICC IIIC	37	17.13	12	8.39		23	21.90	12	11.43	
Grading					0.65					1.00
G1/G2	4	1.85	1	0.70		2	1.90	1	0.95	
G3/G4	212	98.15	142	99.30		103	98.10	104	99.05	
R					0.59					1.00
R0	173	80.09	118	82.52		81	77.14	82	78.10	
R1	43	19.91	25	17.48		24	22.86	23	21.90	
Tumor regression grade										
Becker Ia/Ib			22	15.38				9	8.57	
Becker II			37	25.87				25	23.81	
Becker III			84	58.74				71	67.62	

Legend: cT1 = Mucosa/Submucosa; cT2 = Muscularis propria; cT3 = Serosa; cT4 = Adjacent organs; cN0 = no lymph nodemetastasis detected during staging, cN+ = locoregional lymph node metastasis evident during staging; CD = Clavien Dindo Classification; Staging according to UICC 8th edition; *p*-values printed in bold are considered statistically significant.

**Table 3 cancers-13-00290-t003:** Baseline characteristics for patients with mixed Lauren subtype before and after PSM.

	Mixed Subtype (n = 187), Unmatched					Mixed Subtype (n = 112) PS-Matched				
	Surgery Only (n = 116)		CTX + Surgery (n = 71)		*p*-Value	Surgery Only (n = 56)		CTX + Surgery (n = 56)		*p*-Value
	n	%	n	%		n	%	n	%	
Gender					0.87					0.84
Female	40	34.48	23	32.39		19	33.93	17	30.36	
Male	76	65.52	48	67.61		37	66.07	39	69.64	
Age					**<0.0001**					0.79
<70	69	59.48	62	87.32		49	87.50	47	83.93	
>70	47	40.52	9	12.68		7	12.50	9	16.07	
Localization					**0.04**					0.21
Proximal	55	47.41	40	56.34		28	50.00	26	46.43	
Middle	23	19.83	20	28.17		12	21.43	20	35.71	
Distal	36	31.03	9	12.68		15	26.79	8	14.29	
Total	2	1.72	2	2.82		1	1.79	2	3.57	
Clinical Staging					<0.0001					0.93
cT2 cN+/cNx	39	33.62	10	14.08		11	19.64	10	17.86	
cT3/cT4 cN0	10	8.62	8	11.27		5	8.93	6	10.71	
cT3/cT4 cN+/cNx	67	57.76	53	74.65		40	71.43	40	71.43	
Dissected LN (Median)	34 (7–83)		30 (11–68)		**0.02**	33 (7–83)		30 (11–60)		0.19
≤25	24	20.69	24	33.80	**0.06**	13	23.21	20	35.71	0.21
>25	92	79.31	47	66.20		43	76.79	36	64.29	
Complications					0.89					0.81
None	84	72.41	51	71.83		43	76.79	40	71.43	
CD I/II	17	14.66	12	16.90		8	14.29	10	17.86	
CD III-V	15	12.93	8	11.27		5	8.93	6	10.71	
pT					0.17					0.41
pT0/is	0	0.00	3	4.23		0	0.00	3	5.36	
pT1a	2	1.72	0	0.00		0	0.00	0	0.00	
pT1b	11	9.48	7	9.86		5	8.93	6	10.71	
pT2	15	12.93	12	16.90		6	10.71	9	16.07	
pT3	43	37.07	30	42.25		24	42.86	24	42.86	
pT4a	40.00	34.48	16	22.54		18	32.14	12	21.43	
pT4b	5	4.31	3	4.23		3	5.36	2	3.57	
pN					**0.02**					**0.03**
pN0	27	23.28	29	40.85		10	17.86	23	41.07	
pN1	16	13.79	14	19.72		10	17.86	13	23.21	
pN2	23	19.83	5	7.04		11	19.64	5	8.93	
pN3a	35	30.17	17	23.94		17	30.36	12	21.43	
pN3b	15	12.93	6	8.45		8	14.29	3	5.36	
UICC					0.15					0.19
UICC 0	0	0.00	3	4.23		0	0.00	3	5.36	
UICC IA	8	6.90	7	9.86		2	3.57	6	10.71	
UICC IB	8	6.90	7	9.86		2	3.57	5	8.93	
UICC IIA	9	7.76	10	14.08		6	10.71	8	14.29	
UICC IIB	16	13.79	12	16.90		9	16.07	10	17.86	
UICC IIIA	26	22.41	11	15.49		13	23.21	10	17.86	
UICC IIIB	34	29.31	15	21.13		17	30.36	11	19.64	
UICC IIIC	15	12.93	6	8.45		7	12.50	3	5.36	
Grading					0.32					1
G1/G2	8	6.90	2	2.82		3	5.36	2	3.57	
G3/G4	108	93.10	69	97.18		53	94.64	54	96.43	
R										0.78
R0	100	86.21	61	85.92	1.00	48	85.71	50	89.29	
R1	16	13.79	10	14.08		8	14.29	6	10.71	
Tumor regression grade										
Becker Ia/Ib			14	19.72				12	21.43	
Becker II			21	29.58				18	32.14	
Becker III			36	50.70				26	46.43	

Legend: cT1 = Mucosa/Submucosa; cT2 = Muscularis propria; cT3 = Serosa; cT4 = Adjacent organs; cN0 = no lymph nodemetastasis detected during staging, cN+ = locoregional lymph node metastasis evident during staging; CD = Clavien Dindo Classification; Staging according to UICC 8th edition; *p*-values printed in bold are considered statistically significant.

**Table 4 cancers-13-00290-t004:** Univariate and Multivariate regression analysis for overall survival (OS).

Univariate	HR	CI 95%	*p*	Multivariate	HR	CI 95%	*p*
Age > 70	1.36	1.16–1.60	**<0.001**	Age > 70	1.46	1.24–1.73	**<0.001**
Gender ^!^	1.06	0.90–1.25	0.48	Gender ^!^	1.06	0.89–1.26	0.50
Proximal ^$^	1.00		**<0.001**	Proximal ^$^	1.00		**<0.001**
Middle ^$^	0.59	0.47–0.74	**<0.001**	Middle ^$^	0.56	0.44–0.72	**<0.001**
Distal ^$^	0.67	0.54–0.82	**<0.001**	Distal ^$^	0.79	0.63–0.98	**0.03**
Whole ^$^	1.59	1.11–2.27	**0.01**	Whole ^$^	1.03	0.72–1.48	0.88
nCTx	0.88	0.75–1.03	0.11	nCTx	0.84	0.71–1.00	**0.05**
UICC I ^$^	1.00		**<0.001**	UICC I ^$^	1.00		**<0.001**
UICC II ^$^	2.39	1.77–3.23	**<0.001**	UICC II ^$^	2.26	1.67–3.07	**<0.001**
UICC III ^$^	5.23	3.93–6.92	**<0.001**	UICC III ^$^	4.82	3.59–6.47	**<0.001**
Lauren intestinal ^$^	1.00		**0.016**	Lauren intestinal	1.00		**<0.001**
Lauren diffuse ^$^	1.27	1.08–1.51	**0.005**	Lauren diffuse ^$^	1.40	1.15–1.72	**<0.001**
Lauren mixed ^$^	1.19	0.94–1.49	0.143	Lauren mixed ^$^	1.29	1.01–1.65	**0.04**
Number of LN dissected	1.06	0.89–1.27	0.51	Number of LN dissected	0.91	0.76–1.09	0.31
pR0	1.00						
pR1	2.49	2.01–3.09	**<0.001**	pR	1.55	1.23–1.94	**<0.001**
Grading (G1/2 vs. G3/4)	1.23	1.03–1.47	**0.03**	Grading (G1/2 vs. G3/4)	0.98	0.80–1.21	0.85
Clavien Dindo 0 ^$^	1.00		**<0.001**	Clavien Dindo 0 ^$^	1.00		**<0.001**
Grade I/II ^$^	1.29	1.04–1.59	**0.02**	Clavien Dindo I/II ^$^	1.27	1.03–1.56	**0.03**
Grade III/IV ^$^	1.66	1.32–2.09	**<0.001**	Clavien Dindo III/IV ^$^	1.47	1.16–1.86	**<0.001**

Legend: HR = Hazard Ratio, CI95% lower: 95% Confidence Interval lower boundary, CI95% upper: 95% Confidence Interval upper boundary, *p* = *p*-value, ^!^ male vs. female; ^$^ categorical variable, first value is reference (=1.00): Localization, UICC-stage, Lauren subtype, Clavien Dindo grade; *p*-values printed in bold are considered statistically significant.

## Data Availability

The data presented in this study are available on request from the corresponding author. The data are not publicly available due to European data protection regulation.

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
