# Peer review of "Significance of Lauren Classification in Patients Undergoing Neoadjuvant/Perioperative Chemotherapy for Locally Advanced Gastric or Gastroesophageal Junction Cancers—Analysis from a Large Single Center Cohort in Germany"

_cancers, 2021, doi:10.3390/cancers13020290_

Round 1
Reviewer 1 Report
Authors tried to described the usefullness of Lauren classification from single institution data. As authors conclude that the Lauren classification as a factor to be incorporated in MDT decisions in order to avoid ineffective and potentially harmful treatment. I agree in some part of their conclusion . However, there are some points need to be clarified.It could be possible to conclude that neoadjuvant chemotherapy can be avoided for resectable diffuse type gastric cancer.
- Should we avoid chemotherapy for diffuse type operative gastric cancer?
- Shoukd we avoid surgery for diffuse type gastric cancer who responded to chemotherapy since OS is only 21 month?
- Were there any possibility that diffuse type patients who responded to chemotherapy were patients who experienced downstage
- Is there any biomarker to predict who will respond in intestinal types?
Authors should mention these questions in discussion.
Author Response
These are important questions these data cannot answer sufficiently. As long as no prospective data is available on this matter, definitive conclusions are extremely difficult to take. Nonetheless this analysis hints to the fact that avoiding chemotherapy ahead of surgery may be an appropriate way. Avoidance of surgery should not be recommended at all, as this would be the only (admittedly marginal) option of cure. In the present analysis no real downstaging was detected. In the pretherapeutic staging there were 14 patients staged cT1/cT2 ahead of therapy and 7 patients were pT1/pT2, 91 patients staged cT3/cT4 before chemotherapy+surgery and 98 patients were pT3/pT4 afterwards. 12 patients staged cN0 before and 29 patients staged pN0 after therapy whereas 93 were cN+ before and 76 were pN1-3 after. Clinical UICC stages I and II were detected in 14 patients and pathological UICC stages were found in 35 cases. Clinical UICC III stages were found in 79 patients and 70 patients were detected as pathological UICC III. These data indicate a certain downstaging effect, especially in lymph nodes, however this is very difficult to evaluate as pretherapeutic staging modalities have limited ability to predict true pT-/pN- and UICC stages. To the authors knowledge there is no known biomarker to predict histopathologic response in intestinal types. Nonetheless evaluating MSI status might be the only option to determine if patients might benefit from neoadjuvant/perioperative chemotherapy.
These aspects were discussed and added to the Discussion section accordingly (p13, lines 276-283).
Reviewer 2 Report
I commend the authors for their effort on such a large retrospective analysis of resectable gastric cancer patients. The reason for conducting the study is appropriately outlined (i.e. the interaction of Lauren classification and response to neoadjuvant chemotherapy) and has not been at least to my knowledge evaluated in a large cohort of patients (although as authors stated evaluated in prior studies but using various histological classifications, making interpretation difficult).
However, my comments about this manuscript and suggestions for clarification/improvement are as follows:
- While histological classification is important, I don't think that we should/can conclude to solely use this as the main decision for treatment decisions given now the plethora of other molecular determinants being actively investigated. For example, a large meta-analysis recently demonstrated that MSI-H status was not only prognostic marker of gastric cancer but also predictive marker to chemotherapy in resectable gastric cancer (Pietrantonia F et al. J Clin Oncol 2019; 37:3392-3400). MSI-H can occur independently regardless of the histological subtype. Also, there has been much work on the TCGA subtypes of gastric cancer. Is there any data if these patients had MSI status checked that could have influenced the results? If not, then this needs to be stated as a limitation.
- To the first point, I think the interesting aspect of this study was looking at the impact on overall survival of the responders/poor responders (based on the Becker regression grade classification), as the impact of pathological regression on outcomes remains conflicting in the literature.
- However, to point #2, I would be cautious in making definitive conclusions, particularly in Figure 3 where it is suggested that diffuse subtype patients who receive chemotherapy that have tumor regression have significantly worse OS as compared to diffuse, non-responders and diffuse receiving surgery alone as there were only 9 patients who had tumor regression with chemotherapy. It is hard to make any conclusions from such a small number of patients. Is there any confounding factors why these 9 pts may have done worse in this retrospective analysis?
- While we know from the large perioperative trials (e.g. MAGIC, FLOT4-AIO) that ~50% of patients don't get the post-operative/adjuvant treatment and that survival benefit is still seen with neoadjuvant therapy alone, is there any information about the number of patients who did get adjuvant therapy in this study (including those who got surgery alone) and if so, if this had any influence on the results of your study?
- While R2 resected patients were excluded, ~10-20% of patients had R1 resection. I saw no multivariate analysis done for the R1 patients. Did this at all influence OS for any histological subtypes?
- All the analyses was looking at OS which takes into account survival time of patients who developed recurrence and then received subsequent therapies for metastatic disease. Is there any information on disease free survival (DFS) or proportion of patients who recurred and received subsequent therapies for advanced disease in each cohort? This would provide more information about the role of neoadjuvant chemotherapy which is really to treat/prevent microscopic metastatic disease.
- Based on the above and the retrospective analysis, I would be cautious in making a definitive conclusion that we should use diffuse subtype as the sole determinant for deciding whether we give neoadjuvant chemotherapy or not. Furthermore, the field is moving very rapidly where combination regimens of chemo + other agents (e.g. immune checkpoint inhibitors) are being evaluated in the neoadjuvant setting, and these studies may also provide further insight of not only the Lauren classification but also other molecular markers that determine response to neoadjuvant therapy. As such, I would conclude that the analysis from this study suggests that patients with diffuse subtype may not benefit from neoadjuvant chemotherapy but that further exploration of other factors (e.g. molecular markers), validation in prospective studies, and evaluation of other novel treatments are needed.
- Also, the tables should be reformatted. A couple of tables have rows not in the same line as their labels or the columns are too small and the numbers are crunched up. Table 4 is also a bit confusing and would recommend re-designing to two columns for Univariate and Multivariate with then the values labeled with conventional of HR (95% CI) (i.e. eliminate the lower and upper labels and just write 1.3 (1.16 - 1.60).
- There are several instances in the manuscript where acronyms are not defined, and the reader has to make inferences, which is distracting. For example, in the results section, it discusses five-year survival rate and then all of sudden FYSR is used frequently without having been defined. The convention would have been to write "five-year survival rate (FYSR)" at the first mention of this in the paper; once defined, then FYSR can be used after that.
Reviewer 3 Report
Your study is overall good, the analysis and presentation are excellent. The scientific value is not high - the diffuse type carries worse outcome and prognosis anyway and your results follow the already known fact (which is not bad, thinking that the opposite would be very difficult to take). The heterogeneity of the chemo plays against you, but in research I accept that we use what we have. On the other hand it is important to prove, through research, as amny ideas as we can. Good luck.
Author Response
We kindly thank the reviewer for the constructive review. From the authors´point of view the novel finding of this analysis is that ineffective chemotherapy is not evident only in signet ring cell/poorly cohesive cancers but in also in the other subtypes which are summarized under the definition of Lauren diffuse types. We admit that this adds only little to the scientific value, but may be relevant for future research anyway.
Round 2
Reviewer 1 Report
All question raised by reviewer are answered appropriately. It is worth publishing.
Author Response
We kindly thank the reviewer for the positive review of our revised manuscript.
Reviewer 2 Report
I very much appreciate the thoughtful and detailed responses that the authors provided to my comments of the manuscript. They have appropriately addressed all of the scientific concerns with appropriate revisions made to the manuscript addressing these.
My final comments are very minor English/grammatical and organizational revisions to the additions to the end of the Discussion section:
Recommend making more concise/rewording as follows:
"Other molecular markers, including microsattelite instability (MSI-H) and the Cancer Genome Atlas (TCGA) molecular subtypes (reference the TCGA gastric paper), that could have influenced the results of our study were not assessed and is a limitation of our study. A recently published meta-analysis demonstrated that MSI-H could predict outcomes to neoadjuvant chemotherapy (Pietrantonia F et al. J Clin Oncol 2019; 37:3392-3400). However, this same meta-analysis revealed that a very small proportion (2.4%) of non-intestinal type gastric cancers were MSI-H. In the present analysis the number of patients not responding to chemotherapy in the diffuse type group was markedly higher (> 80% for non-intestinal type cancer) which does not explain the influence of MSI-status only.
Beyond molecular factors, the amount of chemotherapy administered may have also been a confounding factor. Although most of the patients received neoadjuvant chemotherapy only (94.6%), relatively few patients (5.4%) received the perioperative FLOT/MAGIC regimens (i.e. pre-operative + post-operative). Due to incomplete data available about the administration of post-operative chemotherapy, we are unable to determine the influence of post-operative chemotherapy on the outcomes in our study.
Nevertheless, despite these limitations, the results of our study raises questions about the benefit of neoadjuvant/perioperative chemotherapy in diffuse-type gastric cancer. Complete surgical resection remains the only curative option for gastric cancer patients, even if overall survival is markedly shorter for patients with diffuse-type histology as compared to intestinal or mixed type histologies. To the authors´ knowledge, except for possibly MSI status which represents a small proportion of patients, there is no existing biomarker in clinical practice that can adequately predict clinical and histopathologic response to neoadjuvant chemotherapy, and future research on identifying other molecular markers are needed."
Author Response
The reviewers kind recommendation for re-wording was edited in the revised manusript accordingly.